# Visible-Light-Induced Photocatalytic Degradation of Rhodamine B Dye Using a CuS/ZnS p-n Heterojunction Nanocomposite under Visible-Light Irradiation

Rachel Mugumo, Emmanuel Ichipi, Shepherd M. Tichapondwa [ID] and Evans M. Nkhalambayausi Chirwa *[ID]

Water Utilisation and Environmental Engineering Division, Department of Chemical Engineering, University of Pretoria, Pretoria 0028, South Africa; rachel.mugumo@tuks.co.za (R.M.); u17227870@tuks.co.za (E.I.); shepherd.tichapondwa@up.ac.za (S.M.T.)
* Correspondence: evans.chirwa@up.ac.za

**Abstract:** The aim of this work was to investigate a new, simple, one-pot combustion synthesis technique for creating sulphur-based CuS/ZnS p-n heterojunction nanocomposite photocatalysts. This study examined the photocatalytic activity and reusability of these nanocomposites in removing rhodamine B (RhB) dye under visible-light irradiation. Various methods of characterisation were employed to determine the properties of the materials, including particle morphology, crystalline phases, and bandgap energy. The intrinsic reaction parameters, such as catalyst loading, the pH level of the solution, and initial pollutant concentration, were varied to establish the optimal photodegradation conditions. The results showed that a binary CuS/ZnS catalyst with a 10 g L$^{-1}$ loading, at pH 5, degraded 97% of 5 ppm RhB dye after 270 min of visible light irradiation. Additionally, this composite catalyst exhibited excellent chemical stability and reusability, achieving 83% RhB dye removal after five recycling runs. Scavenger tests identified the photogenerated holes (h$^+$) and superoxide free radicals ($\bullet$O$_2$) as the primary reactive species responsible for degradation. This study provides valuable insight into the design of highly efficient nanomaterials for removing organic pollutants in wastewater, and a possible reaction mechanism is proposed.

**Keywords:** CuS/ZnS; photocatalytic degradation; p-n heterojunction; thermal decomposition; one-pot combustion; rhodamine B dye; visible light irradiation

## 1. Introduction

Water contamination by toxic chemicals is a major global environmental concern. The intensive use of organic chemicals and medicines driven by industrial demand, population growth, and agricultural expansion has led to the mass pollution of aquatic systems the world over. A combination of both priority and emerging pollutants are detected at variant concentrations in water bodies [1]. This has led to the development of several treatment technologies capable of remediating these toxic compounds from aquatic environments. Textile effluents are regarded as one of the major water polluters. They typically consist of mixtures of dyes, sulphur, nitrates, soaps, naphthol, acetate acid, chromium, and other heavy metals such as mercury, copper, cobalt, arsenic, and nickel [2]. Some of these chemicals have mutagenic properties and tend to bioaccumulate, resulting in adverse effects on plant, animal, and human life [3]. Generally, dyes make up the largest component of the organic contaminants in industrial effluents. It has been reported that approximately 20% of these effluents consist of synthetic dyes such as rhodamine B, methyl red, Congo red, and methylene blue, which are lost in the dyeing processing steps and discharged into water bodies without further treatment [4].

Rhodamine B (RhB) is a cationic xanthene dye that carries an acidic non-esterified phenyl-carboxylic group [5]. RhB dye is widely used in the textiles, paper, food, and printing industries as a pigment due to its incredible fluorescence pigment which is highly

soluble in water and cost-effective [6]. It is, however, characterised by a non-biodegradable complex chemical structure that is stable to heat and light [7]. RhB dye is a known neurotoxin and carcinogen, which also causes respiratory diseases such as infection, pneumonia, asthma, lung cancer, etc. [8]. It is therefore necessary to reduce the undesirable effects that these dyes pose.

Various conventional techniques have been widely applied in wastewater treatment, including biodegradation, adsorption, flocculation, chlorination, coagulation, and membrane separation processes to treat wastewater effluent [7]. However, these methods have several limitations, such as high sludge generation, high operational costs, long processing times, separation difficulties, and chemical instability [9]. Additionally, they suffer from the stubborn recalcitrant nature of the target pollutants and the generation of secondary pollutants [10]. As a result, it is therefore crucial to develop alternative effective methods for removing organic pollutants from wastewater before discharge into water bodies. This has led to an increased focus on Advanced Oxidation Processes (AOPs) due to their inherent advantages, including operation under ambient conditions, non-selective degradation of organic and microbial pollution, elimination of sludge production, and their sunlight harnessing ability [11]. Advanced Oxidation Processes (AOPs) such as sonolysis, ozonation, the Fenton process, photolysis, and photocatalysis (Table 1) have shown good potential in degrading and mineralising refractory toxic organic pollutants [12]. These AOPs typically function by producing highly reactive, fast-acting, non-selective free radicals which attack the target organic pollutants to form intermediate products or benign $H_2O$ and $CO_2$ [13].

**Table 1.** Various AOP techniques using sulphur-based p-n heterojunction nanocomposites and their photocatalytic performance.

| AOP Technique | S-Based Photocatalyst | Synthesis Method | Target Pollutant | Photocatalytic Activity | Reference |
|---|---|---|---|---|---|
| Photo-Fenton | $MoS_2/Mo_2N$ | hydrothermal | 20 mg $L^{-1}$ paracetamols | 71% degradation in 120 min under UV light | [14] |
| Electrochemical | $SA/TiO_2$ | immersion | 5 mg $L^{-1}$ p-nitrophenol | 5% degradation in 120 min under UV light and 5% in 180 min under visible light | [15] |
| Ozonation | $Fe_2O_3/S\text{-}C_3N_4$ | one-pot in situ | 10 mg $L^{-1}$ methylene blue (MB) dye | 96% removal in 240 min under UV light | [16] |
| Sonolysis | $Cu_xS/ZnO/TiO_2$ | spray pyrolysis | 0.58 mg $L^{-1}$ phenol | 72% degradation in 600 min under UV light | [17] |
| Photocatalysis | $CoS_2/MoS_2$ | hydrothermal | 10 mg $L^{-1}$ rhodamine B (RhB) dye | 78% degradation in 60 min under visible light | [18] |
| Photocatalysis | g-$C_3N_4(SCN)/TiO_2$ | electrospinning | 50 mg $L^{-1}$ congo red (CR) dye | 100% degradation in 60 min under UV light | [19] |
| Photocatalysis | $S@GO/TiO_2$ | ultrasonication | 0.3 mg $L^{-1}$ methylene blue (MB) dye | 93% degradation in 120 min under UV light | [20] |

Photocatalysis is one of the most researched AOPs due to its high degradation efficiency and continuously evolving nature, and it being regarded as a green, sustainable water treatment technology [8]. However, the application of common semiconductor photocatalysts such as $TiO_2$ and ZnO is limited by their wide bandgap energy, which makes them active under UV light irradiation only [21]. Other significant drawbacks include a high recombination rate of the photogenerated electron and hole pairs ($e^-/h^+$) and poor chemical stability for recycling purposes [22]. Bandgap engineering to tailor semiconductor materials for inhibition of the recombination rate and visible-light activation is a new concept in photocatalytic applications [23]. This involves various strategies such as metal doping and heterojunction formation of composite catalysts to enhance their photocatalytic activity under visible-light irradiation [24]. This concept has proven to be effective in inhibiting $e^-/h^+$ pair recombination, which in turn improves the photocatalytic activity of the material [25]. Heterojunction-based catalysts also promote and accelerate the mi-

gration of $e^-/h^+$ pairs, resulting in a more efficient process. Typically, p-n heterojunction semiconductor systems are used in photocatalysis [26]. The photons with energies equal to or higher than the semiconductor photocatalyst bandgap energy create a built-in electric field within the space-charge region, which quickly separates the photoinduced $e^-/h^+$ pairs during irradiation. This electric field drives the transfer of electrons to the conduction band ($C_B$) of n-type and holes to the valence band ($V_B$) of p-type semiconductors. Some of the advantages of p-n type heterostructures include (i) rapid charge transfer of catalysts; (ii) effective charge separation; (iii) longer charge carrier lifetimes; and (iv) separation of incompatible redox reactions [27].

ZnS and CuS are notable examples of sulphur-based semiconductor photocatalysts due to their eco-friendliness, affordability, low toxicity, and exceptional photo-absorption capabilities [28]. ZnS has a bandgap energy of around 3.7 eV, which is relatively wide, and can only absorb about 4% of total sunlight in the UV range. Additionally, it exhibits rapid recombination rates for photoinduced charge carriers, which limits its practicality as a photocatalyst. On the other hand, although CuS has a narrow bandgap of about 2.2 eV, it is prone to photocorrosion. This drawback can be effectively resolved by coupling CuS with ZnS [29]. This results in the formation of a p-n heterojunction photocatalyst with superior photoactivity under visible-light irradiation [30]. The CuS/ZnS heterojunction nanoparticles are characterised by heterogeneous interfaces and an increased number of lattice defects. These defects introduce vacancies in the crystal lattice, creating impurity states within the bandgap. This enhances the ability of the catalyst to utilise visible light and additionally plays a crucial role in facilitating electron transfer by serving as an electron trap, thereby inhibiting the recombination of photogenerated carriers [31]. Moreover, the partitioning of sulphur dopants within the CuS/ZnS p-n heterojunction nanocomposite and the presence of lattice defects expose more active sites, leading to a further enhancement in the photocatalytic activity of the catalyst and improved recyclability [32]. Mondal et al. [30] reported highly improved methylene blue removal through the coupling of binary p-n heterojunction CuS/ZnS photocatalysts, unlike CuS and ZnS prepared via the ion-exchange hydrothermal method. Another study by Sitinjak et al. [33] demonstrated the dramatic enhancement of 4-aminophenol degradation under visible light using the binary CuS/ZnS composite (100%), yet 64% and 28% degradation was reported for the individual constituent CuS and ZnS catalysts, respectively.

A binary p-n CuS/ZnS heterojunction photocatalyst was synthesised using an innovative facile combustion method, which was also applied in the synthesis of the constituent pristine CuS and ZnS nanocomposites. The structure, shape, and optical properties of the as-prepared CuS/ZnS heterojunction were characterized. Furthermore, the photocatalytic activity of the catalysts was evaluated by degrading rhodamine B in aqueous media. A mechanistic photocatalytic degradation mechanism was also proposed.

## 2. Results and Discussion

### 2.1. XRD Analysis

Figure 1 shows the XRD patterns of the prepared ZnS, CuS, and CuS/ZnS nanoparticles. The ZnS XRD diffractogram exhibits diffraction peaks at 32°, 47°, and 56° corresponding to the (102), (112), and (205) planes of the cubic ZnS phase (JCPDS card no: 89-2194). CuS possessed peaks at 29°, 31°, 47°, and 58° corresponding to the (016), (103), (107), and (116) planes of the hexagonal CuS phase (JCPDS card no: 24-0060). The noted diffraction peaks between 12° and 29° and at 80° (represented by asterisks) in the CuS/ZnS nanocomposite corresponding to the (100), (101), and (102) planes (JCPDS card no: 78-0876) can be identified as characteristic CuS peaks. It is noteworthy that the characteristic peaks of pristine ZnS and the CuS/ZnS composite catalyst were almost identical. This was attributed to the low concentrations of the CuS, which is the dopant material used in the synthesis of the binary composite. The CuS nanoparticles were highly amorphous with smeared broad diffraction peaks caused by small crystalline sizes estimated to be about 11 nm. The broadened CuS peaks and reduced intensities suggest a decline in particle size

and crystallinity [29]. The CuS/ZnS spectra reveal a combination of both CuS and ZnS nanomaterials, indicating the successful formation of the CuS/ZnS nanocomposite [30]. The characteristic distinct diffraction peaks of the materials confirmed the crystalline nature and the relative purity of the synthesised catalysts. The average crystallite size (D) of the nanomaterials was calculated using the Debye–Scherrer equation.

$$D = \frac{k\lambda}{\cos\theta} \tag{1}$$

where k is a constant (0.96), λ is the wavelength of the X-ray (0.15418 nm) Cu Kα radiation (1.5406 Å), β denotes the full width at half maximum (FWHM) in radians of the specific peaks, and θ designates the Braggs diffraction angle in degrees. The average crystallite sizes of the ZnS, CuS, and CuS/ZnS nanocomposites are approximately 14 nm, 7 nm, and 17 nm, respectively. The broadening observed in the diffraction peaks, particularly the CuS spectra, is attributed to the small crystallite size [34]. Additionally, an increase in the diffraction peak intensity implies a decrease in FWHM and vice versa. This produces crystal defects in the dopant region and, consequently, a generation of charge imbalance [35].

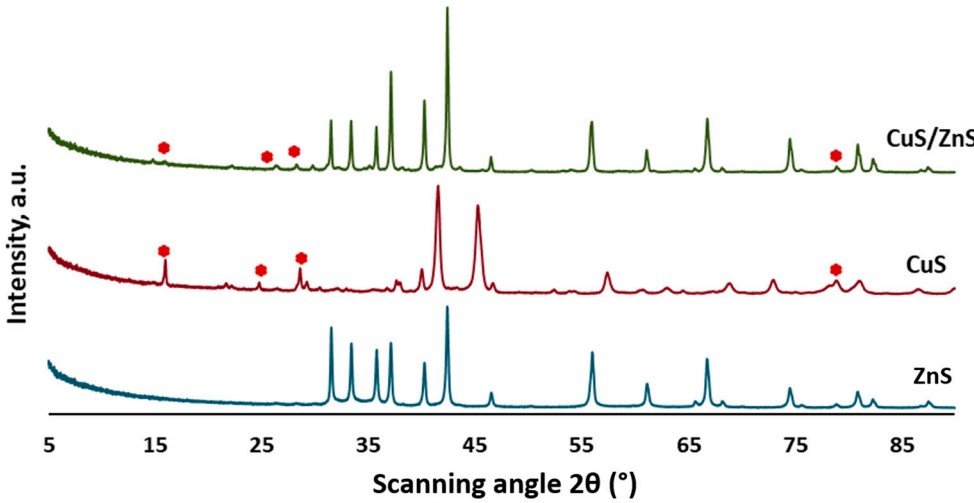

**Figure 1.** XRD patterns of the pristine ZnS and CuS as well as the CuS/ZnS composite.

## 2.2. Catalyst Morphology

Scanning electron microscopy was used to capture and analyse the morphologies of the synthesised ZnS, CuS, and CuS/ZnS as illustrated in Figure 2. The insets in Figure 2a–c depict the synthesised ZnS, CuS, and CuS/ZnS nanomaterials as mixtures of porous agglomerates covering their surfaces with wide particle size distribution. The agglomeration evidenced in all the materials was likely due to high-temperature annealing conditions used during catalysts synthesis [30]. The TEM images presented in Figure 2d–f confirm that these agglomerates were in fact composed of a mixture of nanosphere- and nanoplatelet-shaped primary particles. The captured TEM images further reveal successful CuS/ZnS heterojunction formation as the CuS nanospheres were auspiciously embedded in the ZnS nanoplatelet nanostructures. It has been reported that particle morphology plays an important role in photocatalytic activity as it influences the rate of electron–hole recombination [36]. Additionally, the nanosized nature of the particles enhances the specific surface area of the catalysts, which in turn affects the photodegradation efficacy [37].

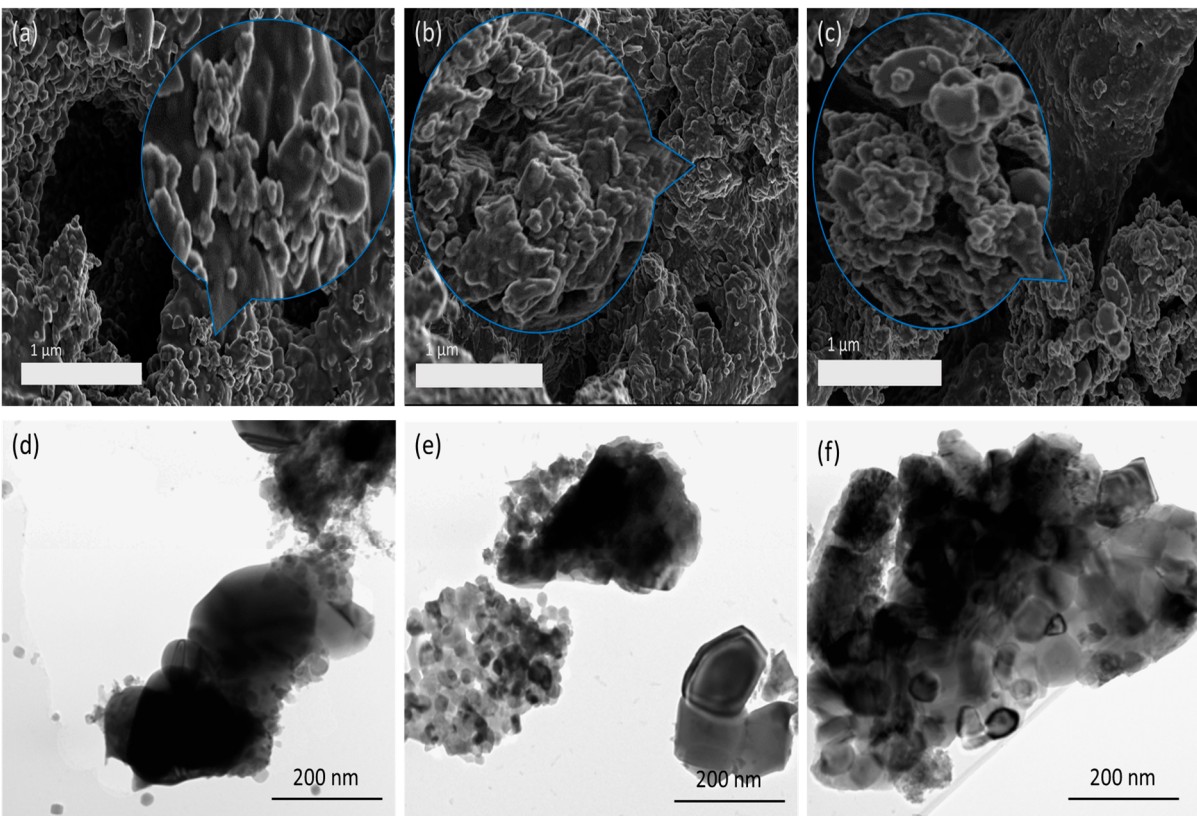

**Figure 2.** SEM images of (**a**) ZnS, (**b**) CuS, and (**c**) CuS/ZnS, and TEM images of (**d**) ZnS, (**e**) CuS, and (**f**) CuS/ZnS.

### 2.3. Surface Area and Pore Size Analysis

The surface area and the pore size distribution of the ZnS, CuS, and CuS/ZnS nanocomposites were analysed using the BET analysis. The results presented in Table 2 further prove that the synthesised materials contained a mixture of mesoporous structures, which supports the mixture of pores observed in the SEM image insets. Large mesopores were observed, recording an average pore size of 12.03 nm (ZnS) and 11.22 nm (CuS); however, smaller mesospheres with an average pore size of 8.57 nm were obtained for the composite CuS/ZnS catalyst. Additionally, the surface areas for the pristine ZnS and CuS were 4.06 m$^2$ g$^{-1}$ and 8.73 m$^2$ g$^{-1}$, while that of the CuS/ZnS was 2.72 m$^2$ g$^{-1}$. Rameshbabu et al. [38] reported similar observations with pure ZnS having a high surface area that decreased upon loading CuS onto the surface of ZnS. According to Hong et al. [29], as the CuS dopant amount increases beyond 3%, the pore size and surface area decrease, resulting in pore blockage when loaded with increased CuS. In the synthesis of the CuS/ZnS binary composite for this study, about 5% of CuS as the dopant material was utilized based on stoichiometric calculations. The decrease could also be a result of the high calcination temperatures used in the combustion synthesis method, and hence particle fusion resulting from the formation of agglomerates as depicted by the catalyst surface characterisation [39]. The broad characteristic peaks observed in the XRD spectra of the constituent materials suggest a decreased crystallinity, which can be attributed to their high surface area and pore size. The specific surface area and porosity of photocatalysts significantly influence the visible-light activity and pollutant absorption of CuS/ZnS, thus improving its overall photocatalytic performance [37].

**Table 2.** Surface area and pore size distribution analysis.

| Materials | Surface Area ($m^2 \, g^{-1}$) | Average Pore Size (nm) | BJH Adsorption (4 V/Å) * | BJH Desorption (4 V/Å) * |
|---|---|---|---|---|
| ZnS | 4.06 | 12.03 | 73.47 | 75.61 |
| CuS | 8.73 | 11.22 | 60.68 | 82.43 |
| CuS/ZnS | 2.72 | 8.57 | 51.35 | 54.38 |

* Average pore width.

Figure 3 depicts the nitrogen adsorption–desorption isotherms of the prepared catalysts. The isotherms of these catalysts can be characterised using the International Union of Pure and Applied Chemistry (IUPAC) classification as Type 4 with H3 hysteresis loops, which confirm the synthesis of mesoporous materials whilst the step-down of the curves represent the spontaneous evaporation of the metastable pore liquid, also known as "cavitation" [40]. The isotherms shown in Figure 3 demonstrate high adsorption levels, particularly in the hysteresis loops observed at high relative pressures ranging between 0.8 and 1.0 p/po. These hysteresis loops are characterized as type H3, which indicates the presence of non-rigid aggregates of plate-like particles giving rise to slit-like pores in the materials [34].

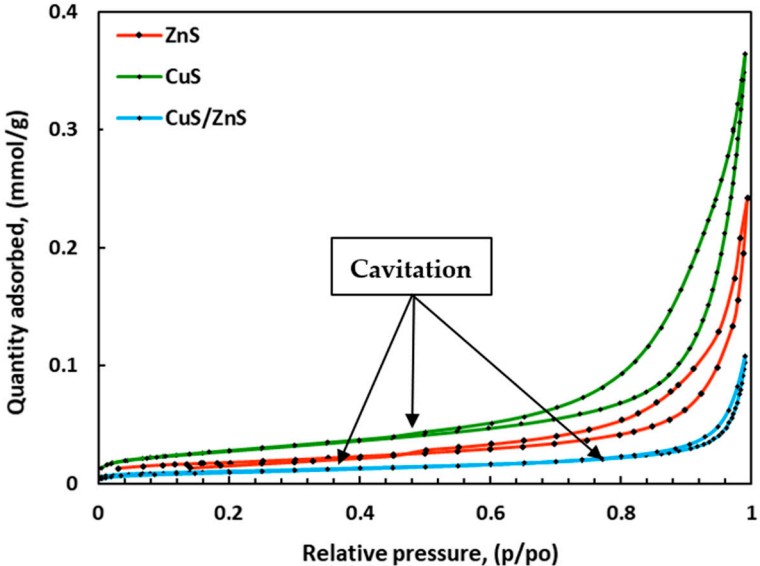

**Figure 3.** $N_2$ adsorption–desorption isotherm of the synthesised CuS/ZnS composite.

### 2.4. Photo-Absorption and Bandgap Analysis

The optical absorption properties of the prepared photocatalysts were analysed by a UV-vis diffuse reflectance spectroscopy (DRS) and the results are shown in Figure 4a. The broad peaks between 339 nm and 378 nm can be attributed to the intrinsic bandgap absorption of ZnS due to transition of electrons from the valence band to the conduction band. The absorption edge for CuS ranges from 300 nm to 800 nm with an infinitesimally slight peak observed at about 588 nm, which is in the visible-light region and can be attributed to $Cu^{2+}$ d-d transition as closely reported by Yu et al. [41], Huang et al. [42], and Rameshbabu et al. [38]. The coupling of minute CuS quantities to ZnS enhanced the visible-light absorption properties of ZnS and extended the visible-light absorption as observed on the distinct absorption edge hump for CuS/ZnS at 380 nm. It can therefore be concluded that the heterojunction formation between ZnS and CuS resulted in increased adsorption in the visible range ~340 nm to 600 nm.

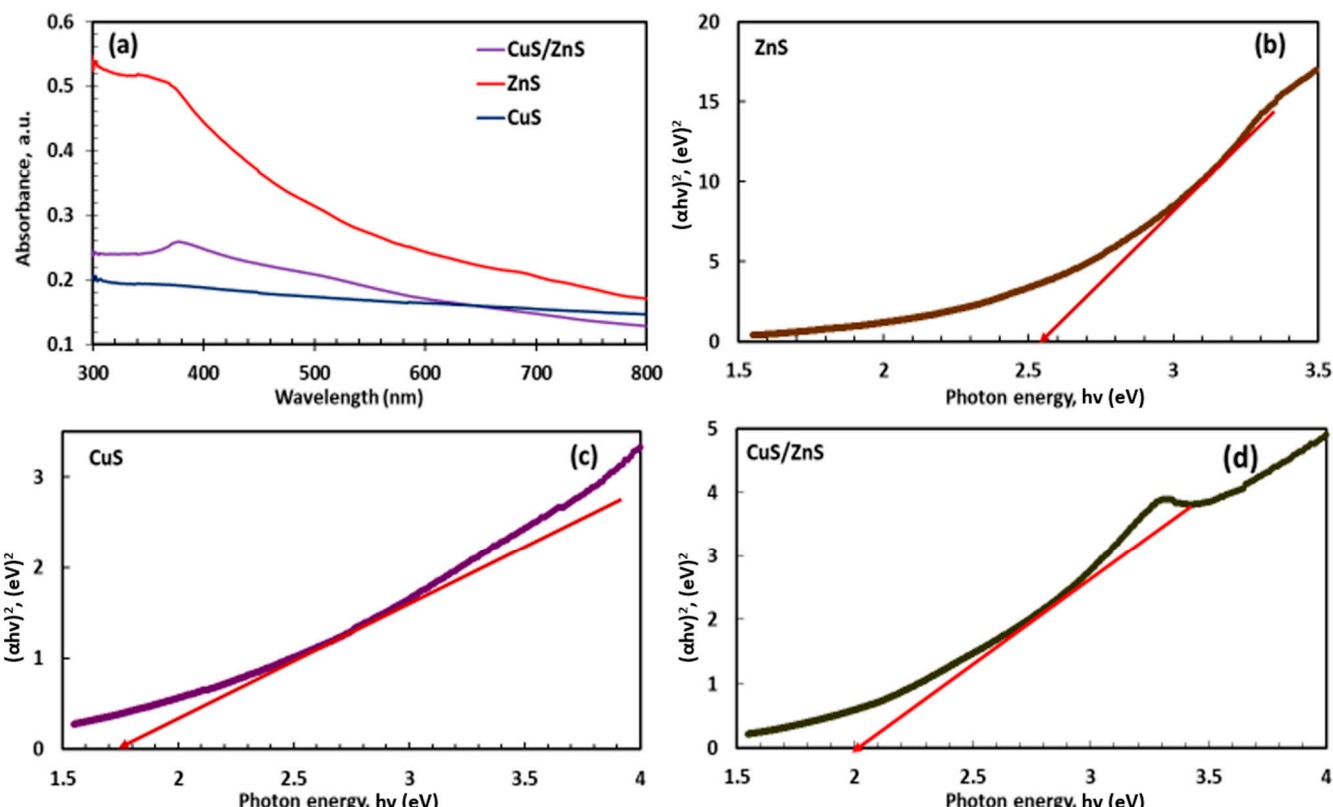

**Figure 4.** (**a**) UV-vis absorption spectra of synthesised nanoparticles, Tauc plot, and estimated bandgap for (**b**) ZnS, (**c**) CuS, and (**d**) CuS/ZnS.

The electronic state and bandgap energies of the synthesised nanocomposites were determined from the obtained spectrum data using the UV-vis DRS differential curves as depicted in Figure 4b–d. The band gap energies of each catalyst material were estimated using absorbance Tauc plots based on Equation (2).

$$(\alpha h v)^n = A(h v - E_g) \tag{2}$$

where $\alpha$ is the absorption coefficient, h is Plank's constant, v is the photon frequency, A is the proportionality constant, hv is photon energy, $E_g$ is the bandgap energy, and n is semiconductor electronic transition where n = 2 for direct allowed transitions and n = 1/2 for indirect allowed transitions. The bandgap energy for each of the catalysts was estimated using plots of $(\alpha h v)^{\frac{1}{2}}$ against $E_g$ as they all had indirect bandgaps.

The bandgap energies of the synthesised ZnS, CuS, and CuS/ZnS were determined to be 2.55 eV, 1.7 eV, and 2.0 eV, respectively. These values are consistent with those reported by Rameshbabu et al. [38] for ZnS at 2.54 eV and CuS at 1.70 eV. The improved visible-light photocatalytic activity of CuS/ZnS can be attributed to the incorporation of $Cu^{2+}$ ions into the ZnS lattice, which leads to the formation of CuS nuclei on the surface of ZnS and the subsequent production of a CuS/ZnS heterojunction. This effect has also been observed by Mondal et al. [30], who reported a reduced bandgap energy from 3.35 eV to 1.99 eV for CuS/ZnS. It is possible that this shift in bandgap energy is a result of the quantum non-confinement effect of CuS/ZnS nanocomposites, as reported by Adelifard et al. [43].

### 2.5. Photocatalytic Performance

The photocatalytic activity of the synthesized catalyst was evaluated by investigating its ability to degrade rhodamine B under visible-light irradiation. Control experiments were performed to assess the impact of visible light alone (photolysis) and the adsorption capacity of the catalyst in the absence of light on rhodamine B dye. The photolysis test

did not result in any degradation, while the adsorption experiment showed significant dye removal, with 67% removed after 4.5 h of contact time. However, the CuS/ZnS photocatalyst exhibited a remarkable degradation efficiency of 97% under photocatalytic conditions with visible-light irradiation for the same duration. These findings (Figure 5a) suggest that the degradation of RhB dye depends on the presence of a photocatalyst and exposure to light, both of which are crucial components of the photocatalytic degradation process. Photocatalysts are capable of absorbing light energy and utilising it to promote surface chemical reactions, whereas light provides the necessary energy to generate reactive electron–hole pairs.

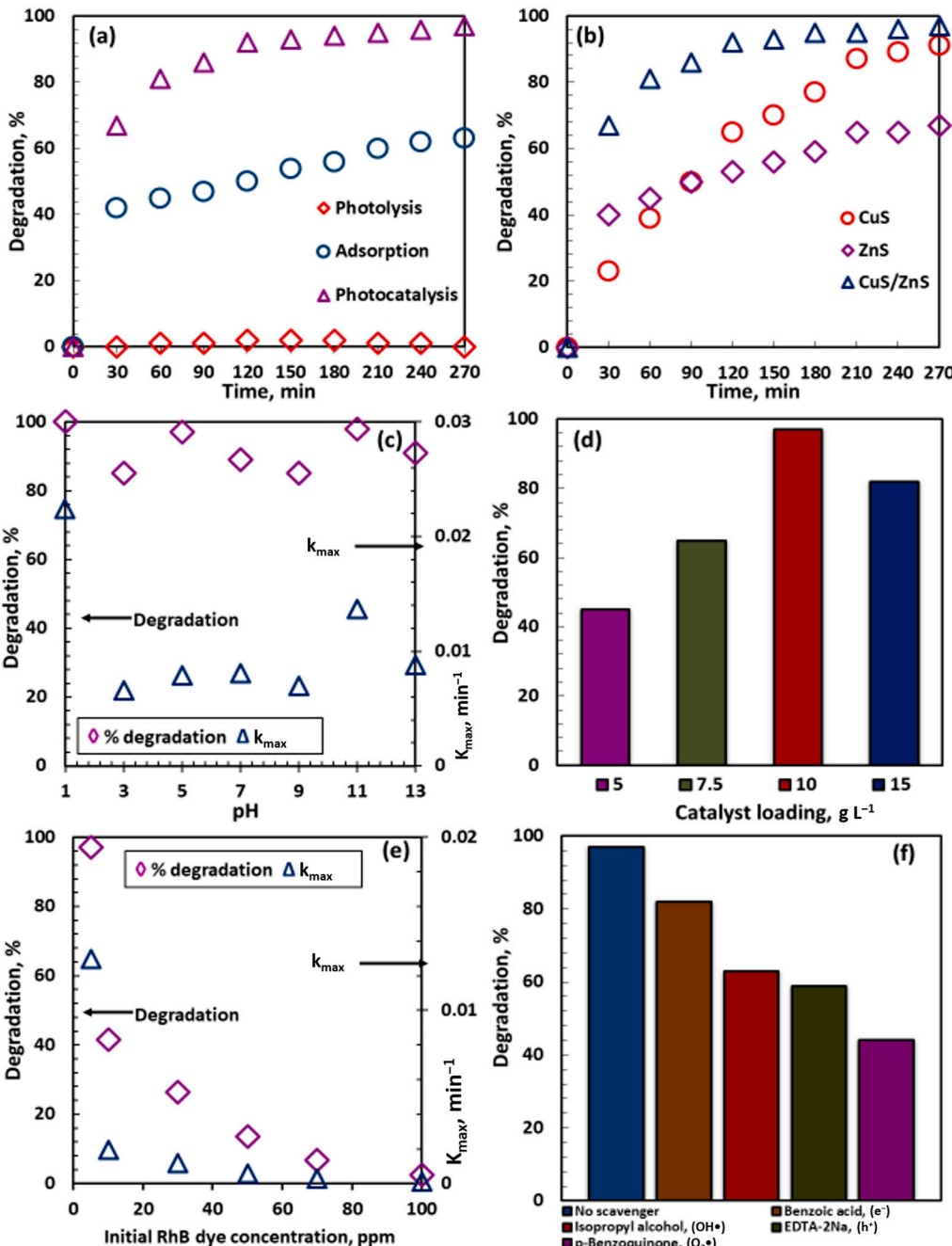

**Figure 5.** (**a**) Control tests and photocatalysis of CuS/ZnS, (**b**) performance of constituent catalyst. Reaction kinetics and effects of (**c**) initial pollutant pH, (**d**) catalyst loading, (**e**) initial rhodamine B solution concentration, and (**f**) radical scavengers in rhodamine B degradation with CuS/ZnS photocatalyst.

### 2.5.1. Effect of Individual Catalyst Composites

The photocatalytic degradation efficiencies of the composite catalysts as well as those of the individual constituents were investigated using 10 g L$^{-1}$ loading of the relevant catalyst on a 5 ppm rhodamine B solution at pH 5. The resulting slurry was stirred continuously for 30 min in the dark in order to establish adsorption–desorption equilibrium prior to a subsequent 4 h of visible-light irradiation. All three catalysts presented fairly high adsorption capacities, with CuS resulting in 23% dye removal after the initial 30 min in the dark which increased to 91% upon visible-light irradiation, followed by ZnS with 40% dye removal, which increased to 67%, and CuS/ZnS adsorbing 67% of the rhodamine B dye, which further increases to 97% upon visible-light irradiation (Figure 5b). The results showed that the constituent materials (CuS and ZnS), however, performed well in the dark as adsorbents due to their surface chemical properties, selective adsorption properties, and high surface area (supported by Table 1). Additionally, the achieved high surface-area-to-volume ratio implies an adsorption of large contaminant (RhB dye) amounts per unit mass. The photocatalytic enhancement of CuS/ZnS can be attributed to several effects: (i) reduced e$^{-}$/h$^{+}$ pair recombination rate because of interface existence between these two phases due to $C_B$ and $V_B$ energy level differences, and (ii) the composite possesses a nanoporous structure that allows for more efficient transport of the reactant molecules of RhB dye onto the available active sites [34]. Harish et al. [28] reported the same phenomena where the photoactivity of the nanomaterials was tested on methylene blue dye under visible light and the results reported 89.70% removal with ZnS, 61.05% with CuS, and a significant 95.51% MB dye degradation after incorporating CuS into the composition in a heterojunction formation.

### 2.5.2. Effect of Initial pH

The solution pH is an important parameter that affects the photocatalytic performance of semiconducting metal sulphides. The surface charge of the photocatalyst and speciation of the target pollutant are influenced by pH, which in turn affects the subsequent adsorption and photocatalytic processes [44]. Figure 5c illustrates the effect of varying the initial solution pH from acidic to basic conditions (pH 1–13). The kinetic analysis was subsequently conducted to further assess the synthesised photocatalysts' degradation proficiency. The Langmuir–Hinshelwood kinetic model expression shown in Equation (3) was used to determine the $k_{max}$ values.

$$k_{max}t = \text{In}\left(\frac{C_0}{C_t}\right) \tag{3}$$

where $k_{max}$ is the pseudo-first-order rate constant, and $C_0$ and $C_t$ are RhB concentrations at t = 0 and time "t", respectively. The dye degradation kinetics are best described using a pseudo-first-order model in the pH range of 3 to 11, with $R^2$ values higher than 0.96 being observed. However, extreme pH values of 1 and 13 did not fit this model well, with $R^2$ values less than 0.84 reported (Figure 5c). The average rate constant was 0.0106 min$^{-1}$, although some outliers at pH 1 and pH 11 were observed around 0.0224 min$^{-1}$ and 0.0137 min$^{-1}$, respectively.

### 2.5.3. Effect of Catalyst Loading

Catalyst dosage has a significant impact on the efficiency of the photocatalytic process. Figure 5d shows how the photocatalytic activity of CuS/ZnS nanocomposites in RhB dye removal varies with different catalyst loadings. The results demonstrate a clear enhancement in the photocatalytic activity of CuS/ZnS with an increase in catalyst loading. The optimum catalyst loading was found to be 10 g L$^{-1}$, beyond which there was a decrease in the photodegradation efficiency. The increase in efficiency with increasing catalyst loading up to the optimum value can be attributed to an increase in the quantity of absorbed photons and the quantity of adsorbed RhB dye molecules. However, beyond the optimum value, particle agglomeration and high suspension turbidity hindered light penetration,

resulting in a reduction in RhB dye removal. This can be attributed to the scarcity and low production of free radical species for the reaction due to the blockage of surface-active sites in the photocatalyst.

The rate constants presented in Table 3 followed a similar trend to the degradation results depicted in Figure 5d. The $k_{max}$ values increased five-fold from 0.0034 min$^{-1}$ for 5 g L$^{-1}$ to 0.0186 min$^{-1}$ for 10 g L$^{-1}$. Increasing the catalyst loading to 15 g L$^{-1}$ decreased the rate constant by half. It is likely that the aforementioned particle agglomeration and shielding effects, which hinder the availability of active sites for reaction, were the cause of this decrease. It is important to note that the Langmuir–Hinshelwood kinetic model best fit the results at a 10 g L$^{-1}$ loading with an R$^2$ value of 0.99, with loadings of 5 g L$^{-1}$ and 15 g L$^{-1}$ recording R$^2$ values of ca. 0.90.

**Table 3.** Photodegradation reaction kinetic parameters for varying binary CuS/ZnS nanocomposite catalyst loading in RhB dye degradation.

| Catalyst Loading (g L$^{-1}$) | Linear Regression (R$^2$) | K$_{max}$ (min$^{-1}$) |
|:---:|:---:|:---:|
| 0 | 0.097 | 0.0000 |
| 5 | 0.897 | 0.0034 |
| 7.5 | 0.947 | 0.0058 |
| 10 | 0.986 | 0.0186 |
| 15 | 0.905 | 0.0094 |

2.5.4. Effect of Initial Rhodamine B Dye Concentration

The initial concentration is a significant factor to consider in photocatalytic performance since an increase in concentration will most likely promote competition between the pollutant molecules and the available photogenerated radical species [33]. In this study, the initial RhB dye concentrations were varied from 5 ppm to 100 ppm while maintaining a 10 g L$^{-1}$ catalyst loading. Figure 5e depicts 97% RhB dye removal at 5 ppm, which drastically declined with increased initial RhB dye concentration, showing 42%, 27%, 14%, 7%, and 3% RhB dye photodegradation performance at initial dye solution concentrations of 10 ppm, 30 ppm, 50 ppm, 70 ppm, and 100 ppm. This expected decrease in degradation efficiency with increasing initial RhB dye concentrations is due to increased RhB dye molecules saturating the CuS/ZnS photocatalyst active sites. Increased pollutant concentrations also result in a more darkened colour that restricts electron–hole separation within the catalyst due to light adsorption by the RhB dye solution before reaching the material surface. Furthermore, this can be postulated to reduce the generation of hydroxyl radicals, which ultimately reduces the photodegradation performance [27].

The highest RhB dye removal and fastest reaction rate of 0.01299 min$^{-1}$ was achieved at an initial pollutant concentration of 5 ppm. The rate constant values decreased with increased RhB dye concentration, reporting a $k_{max}$ value of 0.0001128 min$^{-1}$ at an initial RhB concentration of 100 ppm.

*2.6. Radical Scavenging Test*

Reactive oxygen species (ROS) such as superoxide radicals ($O_2^{-\bullet}$) and hydroxyl radicals (OH$\bullet$) play a vital role in the photodegradation of organic pollutants as they are regarded as strong, non-selective oxidants [44]. In order to determine which reactive species was responsible for dye degradation in this study, radical scavenger tests were conducted. The following scavengers were used to target specific reactive species: benzoic acid (e$^-$), EDTA-2A (h$^+$), IPA (OH$\bullet$), and pBZQ ($O_2^{-\bullet}$). These tests were conducted using the optimum conditions: 10 g L$^{-1}$ CuS/ZnS loading, 5 ppm RhB dye concentration, and initial pH of 5. The results as illustrated in Figure 5f revealed that the superoxide radicals ($O_2^{-\bullet}$) were the dominant reactive species as the addition of p-Benzoquinone resulted in a 53% decrease in dye degradation efficiency. Hydroxyl free radicals and holes (h$^+$) also played a role in the degradation mechanism. The removal of these species from the aqueous media yielded degradation efficiencies of 59% and 63% compared to the

97% removal observed when no scavenger was added. The presence of electrons did not significantly retard the performance of the catalyst. This was somewhat surprising since the formation of superoxide free radicals ($O_2^{-\bullet}$) occurs through the reaction of the photogenerated electrons with dissolved oxygen. These results were similar to those reported in studies by Mohanty et al. [45], who credited positively charged holes and superoxide free radical as the dominant reactive species responsible for the degradation of RhB and malachite green (MG) dye when using g-$C_3N_4$/$Bi_4Ti_3O_{12}$ as a photocatalyst illuminated with visible-light irradiation.

### 2.7. Mechanism

CuS has a narrow bandgap energy (1.7 eV), which makes it highly visible-light-photosensitive unlike ZnS (2.5 eV), which has a wide forbidden bandgap energy. The proposed photocatalytic mechanism for the CuS/ZnS heterostructure involves several steps. Firstly, upon irradiation, visible light (hν) excites CuS/ZnS, generating electrons ($e^-$) and holes ($h^+$). These electrons and holes are then available to participate in subsequent reactions. The electrons generated in the conduction band of CuS are captured by $O_2$ to form superoxide anion radicals, contributing to both direct pollutant degradation and indirect RhB removal via reactions with $H_2O$ to generate OH• radicals. As shown in Figure 6, the flow of electrons from the conduction band ($C_B$) of ZnS to CuS is impeded because of a mismatch between their respective bandgap energy levels. Consequently, the isolated electrons in ZnS do not participate in the reduction of $O_2$, as the photodegradation redox potential is determined by the energy levels of the bandgap [46,47]. Additionally, studies by Hong et al. [29] suggest that the electrons in the CuS $C_B$ may lead to a partial reduction of CuS to produce CuS/$Cu_2S$ nanoparticles that serve as electron traps to enhance the separation and migration of photogenerated carriers [29]. The positively charged holes react with water molecules, generating hydrogen ions ($H^+$) and hydroxyl radicals (OH•) which are powerful oxidants that can react with and degrade organic pollutants. According to the interfacial charge transfer (IFCT) mechanism and the presence of Zn vacancies in both ZnS and at the interface, some of the positively charged holes on the valence band of CuS are transferred to the vacancies of ZnS and the heterojunction. Finally, the hydroxyl radical or superoxide radical anion reacts with the organic pollutant, leading to the formation of oxidised products and water. This reaction represents the degradation of the pollutant and is the ultimate goal of the photocatalytic process.

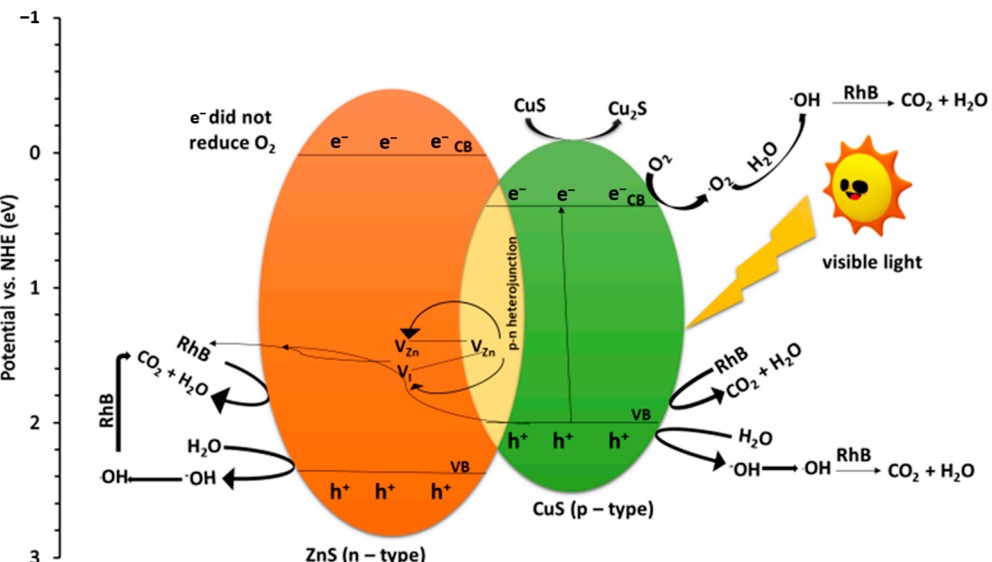

**Figure 6.** Proposed schematic representation of a p-n CuS/ZnS heterojunction photocatalytic mechanism.

This proposed mechanism is similar to the photocatalytic mechanism reported by Feng et al. [46]. The major photocatalytic mechanism steps under visible-light irradiation are summarised by the following equations:

$$\text{Excitation}: CuS/ZnS + hv \rightarrow CuS/ZnS(h^+{}_{VB} + e^-{}_{CB}) \tag{4}$$

$$\text{Recombination}: h^+ + e^- \rightarrow energy \tag{5}$$

$$\text{Oxidation}: h^+{}_{VB} + H_2O \rightarrow OH\bullet + H^+ \tag{6}$$

$$h^+ + OH^- \rightarrow OH\bullet \tag{7}$$

$$OH\bullet + RhB \rightarrow (intermediates) \rightarrow CO_2 + H_2O \tag{8}$$

$$\text{Reduction}: e^-{}_{CB} + O_2 \rightarrow O_2{}^{-\bullet} \tag{9}$$

$$O_2{}^{-\bullet} + H^+ \rightarrow HO_2{}^{\bullet} \tag{10}$$

$$2H_2O + O_2{}^{-\bullet} + e^- \rightarrow OH\bullet + 2OH^- \tag{11}$$

$$OH\bullet/O_2{}^{-\bullet} + RhB \rightarrow (intermediates) \rightarrow CO_2 + H_2O \tag{12}$$

In conclusion, the presence of IFCT in heteronanostructures has been found to enhance photogenerated charge separation, leading to increased photocatalytic activity. The effective separation of photogenerated electrons and holes is achieved through multistep charge transfer at p-n heterojunctions. Moreover, the suggested mechanism is consistent with the radical forensic test conducted (depicted in Figure 5f), which indicated that the process of photodegradation was decelerated upon the addition of p-benzoquinone ($O_2{}^{-\bullet}$ scavenger) and EDTA-2Na ($h^+$ scavenger). This indicates that the degradation mechanism is primarily driven by superoxide radicals and photogenerated holes.

## 3. Catalyst Recyclability

The reusability of photocatalysts is an important factor in determining their cost-effectiveness, as replacements can be quite expensive. To test the stability of the nanocatalyst, five rhodamine B degradation cycles using a recovered catalyst were conducted. After each cycle, the catalyst particles were recovered by centrifugation and then washed with deionised water, ethanol, and deionised water three times prior to thermal reactivation. The catalyst was then oven-dried at 50 °C for 12 h and ground to a powder before reuse. The results depicted in Figure 7 show that the CuS/ZnS nanocatalyst exhibited a high level of photostability, with no significant decrease in photodegradation efficiency after five cycles. Following each cycle, the catalytic activity slightly decreased, resulting in a 14% activity reduction (from 97% in cycle 1 to 83% in cycle 5). It can be concluded that the CuS/ZnS nanomaterial did not lose its degradation properties but demonstrated excellent chemical stability, recyclability utilisation, and sustainability.

Figure 8 shows scanning electron microscope (SEM) images of the CuS/ZnS photocatalyst before and after five cycles of reuse in the photodegradation of the rhodamine B pollutant. The images reveal that the surface of the hollow porous material, with a varied pore size distribution, remained consistent before and after reuse. However, the morphology of the material changed during recycling from clustered agglomerates to a mixture of morphologies, including flower-like particles, nanosheets, and nanoplates. This suggests that the photoactivity of the catalyst used in this study was not greatly influenced by the particle morphology.

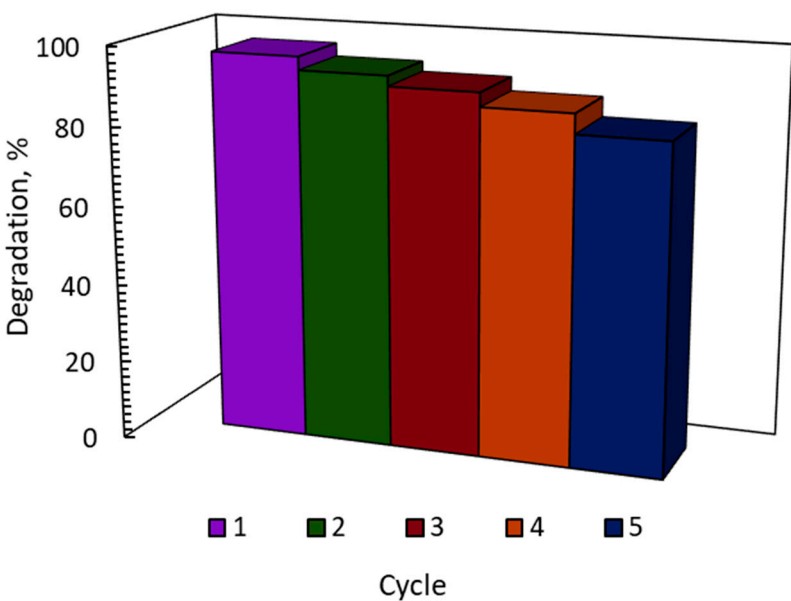

**Figure 7.** Stability and recyclability of CuS/ZnS photocatalyst during 5 cycles.

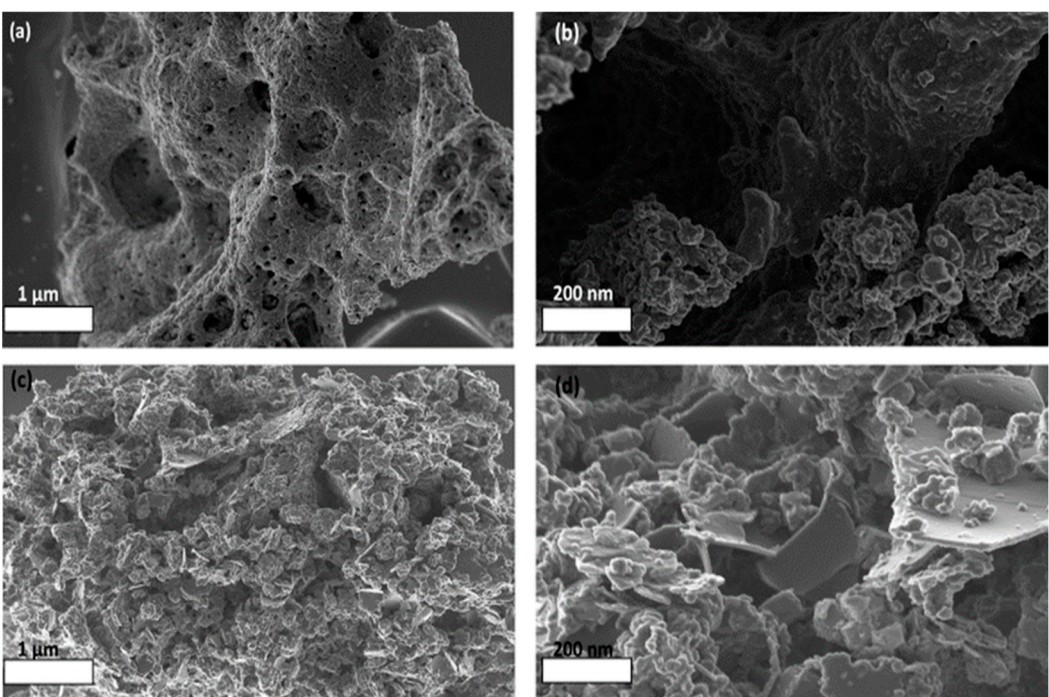

**Figure 8.** (**a**) Zoomed-out and (**b**) zoomed-in CuS/ZnS SEM images before use and (**c**) zoomed-out and (**d**) zoomed-in CuS/ZnS SEM images after 5 recycles.

## 4. Materials and Methods

### 4.1. Chemicals

Copper (II) nitrate trihydrate [Cu(NO$_3$)$_2$·3H$_2$O] (CAS: 10031-43-3) and thiourea [(NH$_2$)$_2$CS] (Batch No: 51378) were purchased from Sigma-Aldrich (St Louis, MO, USA). Zinc (II) nitrate hexahydrate [Zn(NO$_3$)$_2$·6H$_2$O] (CAS: 10196-18-6) was purchased from Glassworld (Johannesburg, South Africa). These served as precursors for the synthesis of the ZnS, CuS, and CuS/ZnS nanocomposites. HPLC-grade rhodamine B (CAS: 81-88-9) was purchased from Sigma-Aldrich, while sodium hydroxide (NaOH, Batch No: SAAR5823200) and nitric acid (HNO$_3$, Batch No: OPCH000065235WO) used in pH adjustments were purchased from

Glassworld. The scavenger tests were conducted using isopropyl alcohol (IPA, Batch No: 19/049) purchased from Glassworld, p-benzoquinone (pBZQ, Batch No: 1421039 55108019) purchased from Sigma-Aldrich, and benzoic acid (BA, Batch No: 1983/008079/07), and EDTA-2Na (Batch No: 83/08079/07) purchased from LabChem (Zelienople, PA, USA). All reagents were used without further purification. Deionised water (DI) was produced by an Elga Purelab Chorus unit purifier.

### 4.2. Catalyst Synthesis

ZnS, CuS, and CuS/ZnS (5% CuS:95% ZnS) nanocomposites were synthesised using a novel and facile one-pot solid-phase method. In this method, stoichiometric amounts of $Zn(NO_3)_2 \cdot 6H_2O$, $(NH_2)_2CS$, and $Cu(NO_3)_2 \cdot 3H_2O$ were weighed out into a crucible using the following ratios 4:1:0.2 before calcining the mixture at 400 °C for 5 h. The resulting product was then ground using a pestle and mortar and sieved through a 25 μm mesh sieve to yield the final powdered photocatalyst. Pristine ZnS and CuS were synthesized in a similar fashion. Zinc (II) nitrate hexahydrate and thiourea were used to produce ZnS while copper (II) nitrate trihydrate and thiourea were mixed to form CuS.

### 4.3. Degradation Studies

The photocatalytic activity of the synthesised nanomaterials was investigated by dispensing predetermined amounts of CuS/ZnS in 100 mL of 5 ppm RhB dye solution. This suspension was continuously stirred in the dark for 30 min in order to attain adsorption–desorption equilibrium prior to subsequent 4 h visible-light irradiation. Aliquot samples of 2 mL were withdrawn every 30 min and centrifuged. The resulting solution was passed through 0.45 μm simplepure filters before analysis. Control photolysis and adsorption tests were also conducted under the same conditions. Optimisation studies were conducted to determine optimum photodegradation conditions while varying CuS/ZnS loading (0–15 g L$^{-1}$), initial RhB dye concentration (5–100 ppm), and initial solution pH (1–13), which was adjusted using 0.1 M $HNO_3$ and 0.1 M NaOH. A WPA, LIGHT Wave, Labotech UV-vis spectrophotometer (Biochrom, Cambridge, UK) was used to analyse the change in RhB dye concentration at a characteristic wavelength of 554 nm using deionised water as a reference (blank). The achieved photodegradation percentage was determined using the following Equation (13):

$$\% \text{ Degradation} = \frac{(C_0 - C_t)}{C_0} \times 100 \tag{13}$$

where $C_0$ is the initial rhodamine B concentration and $C_t$ is rhodamine B concentration after irradiation time, t.

Furthermore, various scavenger tests were conducted to determine the most reactive oxidation species. This was performed through the addition of 5 mmol L$^{-1}$ of benzoic acid (BA for $e^-_{CB}$), isopropyl alcohol (IPA for OH• radicals), ethelene diamino tetra acetylhydride-disodium (EDTA-2Na for $h^+_{VB}$), and p-benzoquinone (p-BZQ for $O_2^{\bullet}$ radicals). Recyclability tests were conducted to investigate the stability of the binary CuS/ZnS. In these tests, a sample was withdrawn after each run for analysis whilst the remaining solution was centrifuged and decanted. The collected catalyst particles were dried at 50 °C overnight and then dispersed into a fresh RhB solution for another run.

### 4.4. Material Characterisation

X-ray diffraction (XRD) spectra of catalysts were analysed using a PANalytical X'Pert Pro powder diffractometer (Malvern Panalytical Ltd., Malvern, UK) in θ–θ configuration with an X'Celerator detector and variable-divergence and fixed receiving slits with Fe-filtered Co-Kα radiation (λ = 1.789 Å). The samples' mineralogy was determined by selecting the best-fitting pattern from the ICSD database to the measured diffraction pattern, using X'Pert Highscore plus software. The scanning electron microscopy (SEM) imaging was captured on a Zeiss Ultra PLUS FEG SEM (Oberkochen, Germany) using the Oxford

instruments detector and Aztec 3.0 software SP1 while the high-resolution transmission electron microscope (HRTEM) images were captured using a JEOL TEM 2100F, 200 kV analytical electron microscope (Tokyo, Japan). A Brunauer–Emmett–Teller (BET) micrometrics Tristar II 3020 Version 3.02 system was used to determine the surface area and pore distribution of the photocatalysts. The samples were degassed overnight at 100 °C prior to analysis. The optical properties of the synthesised nanomaterials were measured using a Hitachi U-3900 (Tokyo, Japan) single monochromatic double-beam UV-vis system which uses UV-solutions software program.

## 5. Conclusions

This study presents a novel and facile one-pot thermal decomposition method for synthesizing CuS, ZnS, and hybrid CuS/ZnS p-n heterojunctions with porous morphologies, which significantly improved their photodegradation efficacy by reducing the rate of charge carrier transfer. The binary CuS/ZnS heterojunction demonstrated exceptional RhB dye photodegradation under visible-light irradiation due to its ability to absorb visible light and the inhibition of the photoinduced $e^-/h^+$ recombination rate. Photodegradation scavenger tests revealed the superoxide radicals to be the primary reactive species responsible for the efficient degradation of RhB dye. Additionally, recycling investigations proved the CuS/ZnS photocatalysts' outstanding durability and excellent chemical stability after five cycles of use. The results of this study highlight the promising potential of the novel synthesis method and resulting CuS/ZnS heterojunction for cost-effective and efficient industrial treatments using photocatalysis. Further studies may explore the applicability of this method to other photocatalytic systems and applications.

**Author Contributions:** Conceptualization, R.M. and S.M.T.; methodology, R.M.; software, R.M.; validation, S.M.T.; formal analysis, R.M.; investigation, R.M.; resources, S.M.T. and E.M.N.C.; data curation, R.M.; writing—original draft preparation, R.M.; writing—review and editing, S.M.T. and E.I.; visualization, R.M.; supervision, S.M.T. and E.M.N.C.; project administration, R.M.; funding acquisition, E.M.N.C. All authors have read and agreed to the published version of the manuscript.

**Funding:** This research was funded by National Research Fund (NRF) of South Africa, grant number EQP180503325881 awarded to Evans M. Nkhalambayausi Chirwa, The NRF Thuthuka Grant grant number TTK18024324064 awarded to Shepherd Tichapondwa and Rand Water Chair in Water Utilisation, grant number RW01413/18. Both Chirwa and Tichapondwa were affiliated to the University of Pretoria during the year of publication of this article.

**Data Availability Statement:** Data will be made available upon request to the corresponding author.

**Conflicts of Interest:** The authors declare no conflict of interest.

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
