# Peer review of "Visible-Light-Induced Photocatalytic Degradation of Rhodamine B Dye Using a CuS/ZnS p-n Heterojunction Nanocomposite under Visible-Light Irradiation"

_catalysts, doi:10.3390/catal13081184_

Round 1
Reviewer 1 Report
The manuscript presents a very interesting study on composites based on CuS and ZnS phases. The experimental scientific approaches are well described, and the results are convincing.
The study by X-ray diffraction can be improved: a refinement of the crystalline parameters could have been carried out and a study of the broadening of Bragg peaks to evaluate the sizes of crystallites is also missing.
Minor revision is needed.
Author Response
We would like to the Reviewer for the valuable comments and recommendations on our original submission. We value the constructive suggestions from Reviewer, which we believe have significantly improved the quality of our article. The revised version of the article is enclosed with detailed responses to Reviewer 1 comments attached below.

Reviewer 2 Report
In this manuscript, the authors reported that one-pot combustion synthesis technique for creating sulphur-based CuS/ZnS p-n heterojunction nanocomposite photocatalysts. The study examined the photocatalytic activity and reusability of these nanocomposites in removing rhodamine B (RhB) dye under visible light irradiation. The authors claimed that this study provides valuable insight into the design of highly efficient nanomaterials for removing organopollutants in wastewater. In overall, this manuscript is interesting but in order to consider publication, this work should be revised. The following comments should be addressed for the improvement of their manuscript.
Comment 1: The overall study aims for creating sulphur-based CuS/ZnS p-n heterojunction nanocomposite photocatalysts and their excellent photocatalytic activity and reusability in wastewater treatment need to be further clarified in detail as compared to current conventional system for organic pollutants degradation and wastewater treatment.
Comment 2: The various recent reports and their research findings on the “sulphur-based CuS/ZnS p-n heterojunction nanocomposite photocatalysts” using various promising techniques and their photocatalytic activity performance in dye degradation under various light illumination should be summarized into a table form and discussed for better understanding in term of benchmarking points with your research findings.
Comment 3: What is the role of lattice defects and element partitioning of sulphur dopants within the CuS/ZnS p-n heterojunction nanocomposite in improving the photocatalysis oxidation reactions? Please discuss and clarify with fundamental support.
Comment 4: . The detailed HRETM and XPS analysis can be included to further explain their lattice fridge, chemical state and electronic state of the elements that exist within sulphur-based CuS/ZnS p-n heterojunction nanocomposite photocatalysts.
Comment 5: In addition, the authors should conduct the PL / EIS spectroscopy analysis to provide information such as charge carrier trapping, immigration, and transfer within sulphur-based CuS/ZnS p-n heterojunction nanocomposite photocatalysts.
Comment 6: The carefully English correction is necessary for the whole manuscript. Please check and revise accordingly.
The carefully English correction is necessary for the whole manuscript. Please check and revise accordingly.
Author Response
Thank you for the in-depth review of our article and for the time you allocated on completing this task. Thanks a lot for the valuable comments that greatly improved the quality and factual presentation of the data and data interpretation. I have included the point by point response to the reviewers queries as indicated in the attached document.

Reviewer 3 Report
This study presents a novel and facile one-pot thermal decomposition method for synthesizing CuS, ZnS, and hybrid CuS/ZnS nanocomposite with porous morphologies, which significantly improved their photodegradation efficacy by reducing the rate of charge carrier transfer. It's better to provide more data and explanation to prove the exist or formation of the binary CuS/ZnS heterojunction.
Author Response
We thank the reviewer for the commendation of this work and offering a chance of publication upon revision. The reviewer suggested that more data be recorded and presented to prove efficacy of the proposed nanocomposite particles. Due the fundamental nature of the research, we decided to present the current data rather than lose the initiative by spending too much time on operational data. The other element that made us skip further characterisation with XPS is due to uniform and stable nature the synthesized particles. We determined that the XPS data yields valuable information on internal structures such as countered in the two-level and three-level z-scheme heterogeneous photocatalysts which is not the case in the current study.

Reviewer 4 Report
The work entitled “Visible-light-induced photocatalytic degradation of Rhodamine B dye using CuS/ZnS p-n heterojunction nanocomposite under visible light irradiation” presents heterostructure synthesis and photocatalytic studies for oxidative degradation of Rhodamine B dye. can be better characterized with some suggestions:
1. It is not clear why a “new” nanocomposite is cited, since this material already has other publications. In the abstract and at other points in the text, emphasize what is new in this study. What is the difference between this work and what has already been published?
2. In the photocatalysis and adsorption studies, the results are somewhat confusing, since the material has a significant substrate adsorption capacity. For example, Figure 5 shows the graphs corresponding to the adsorption and photocatalysis results, but it is not clear how the test was carried out. Separately ? Subsequent processes ? First, a test was carried out with the lamp off for 30 minutes and then with irradiation, but it does not seem that the adsorption/desorption equilibrium is reached in just 30 minutes. I suggest a better description of the results. If there is a combination of adsorption and photobleaching processes and check the total dye removal rate (adsorption + photodegradation). There are publications that display the graph more clearly, here are some examples:
Competitive removal of pharmaceuticals from environmental waters by adsorption and photocatalytic degradation, Environ Sci Pollut Res (2014) 21:11168–11177 DOI 10.1007/s11356-014-2593-5.
Enhanced organic pollutant photodegradation via adsorption/photocatalysis synergy using a 3D g-C3N4/TiO2 free-separation photocatalyst, Chemical Engineering Journal Volume 370, 15 August 2019, Pages 287-294
Highly synergic adsorption/photocatalytic efficiency of Alginate/Bentonite impregnated TiO2 beads for wastewater treatment, Journal of Photochemistry & Photobiology, A: Chemistry 412 (2021) 113215
Author Response
We thank the reviewer for the recommendation of this work and offering us a chance to publish in Catalysts upon revision. We have addressed the Reviewer's comments as indicated in the Point-by-Point commentary below. The reviewer makes a valid point that the Novelty of the results be clearly stated. We have addressed this in the responses accompanying this resubmission.

Round 2
Reviewer 2 Report
In overall, this manuscript was technically well revised. This revised manuscript meets the criteria of Catalysts. Therefore, in my opinion, the revised manuscript can be accepted for publication.